# PRUNE-AS-RECONSTRUCT: MASKED AUTOENCODERS ARE EFFICIENT IMPORTANCE INDICATORS

## ABSTRACT

Network pruning has emerged as an effective technique for reducing the size and computational complexity of neural networks, thereby addressing the challenges of deployment on resource-limited devices. However, existing pruning criteria are predominantly based on handcrafted heuristics or calculated statistics, hindering their generality and effectiveness. In this paper, we reveal that masked autoencoder (MAE) can exploit the hidden semantic information within structured parameters, thereby functioning as a learnable pruning criterion. Specifically, to address the dimension inconsistency problem between layers, we propose a parallel training pipeline, facilitating stable and efficient MAE training on weight matrices. Based on the 'harder-reconstructed-more-important' assumption, we explore diverse pruning strategies and formulate structured pruning as a sample-without-replacement problem that strikes a balance between algorithm complexity and performance. Extensive experiments on benchmark datasets, including CIFAR-10 and ImageNet, demonstrate that our method can efficiently compress both convolutional neural networks and transformers. Furthermore, the trained MAE exhibits transferability across various structures and datasets, avoiding repetitive training from scratch and highlighting its potential as a universal pruning criterion. To the best of our knowledge, this is the first work that establishes a connection between structured pruning and self-supervised learning.

## 1 INTRODUCTION

Deep Neural Networks (DNNs) have revolutionized computer vision in recent years, achieving state-of-the-art performance in a wide range of tasks, including image classification, object detection, and semantic segmentation He et al. (2015); Zhao et al. (2018); Shelhamer et al. (2014). However, the remarkable success of DNNs, whether convolutional neural networks (CNNs) or Vision Transformers (ViTs), comes at the cost of increased computational complexity and memory demands, making it arduous for deployment on resource-constrained devices. To address this challenge, network pruning has emerged as a prominent technique aimed at reducing their size and computational complexity. As explored in prior works Lin et al. (2020); He et al. (2020a); Li et al. (2021), network pruning seeks to identify and remove nonessential parameters while maintaining accuracy.

In the realm of network pruning, structured pruning has gained significant prominence owing to its compatibility with hardware infrastructure. However, existing structured pruning methods typically rely on handcrafted heuristics or calculated statistics to identify the importance of filters. Taking the structured pruning on CNNs for instance, ASFP He et al. (2020b) adopts norm-based criteria for importance evaluation and discards the unimportant weights. Although straightforward, it fails to consider the inter-filter relationships and thus lacks efficiency. ThiNet Luo et al. (2017) measures the reconstruction error between the original feature maps and those generated by the pruned networks, yet it requires iterative data propagation, posing challenges in terms of transferability and deployment. Consequently, these criteria suffer from limited effectiveness and may exhibit poor generalization across diverse tasks and network architectures.

Given the current context, it is crucial to explore a feasible criterion that can provide importance evaluation both spontaneously and without explicit reliance on quantitative data. In this paper, we delve into this issue and propose a novel pruning method based on the masked autoencoder (MAE), dubbed as **MAEP**, to fully exploit the intrinsic correlations among learned parameters.

By connecting the importance evaluation with masked image modeling (MIM), we can uncover the semantic information concealed within structured parameters via MAE. Specifically, we first flatten and pad the weights from each layer to a 2D matrix. The randomly masked 2D matrix is then forward propagated by MAE to reconstruct the original one, where the obtained gradient from each thread is further aggregated to update MAE. With such an efficient training pipeline, we solve the inconsistent dimension issue of different layers, allowing MAE to be trained on the learned parameters of the unpruned network in a batch-wise, parallel manner for faster convergence and better generalization. Once trained, we search for the subsets in each layer that are the most challenging to reconstruct, indicating the most critical ones for preservation. Moreover, leveraging the variable-length input characteristic of MAE, we propose a novel strategy regarding pruning as a sample-without-replacement problem to achieve a better trade-off between performance and complexity. Consequently, the trained MAE can automatically discriminate the importance, either jointly or individually, serving as a better indicator to guide pruning. In contrast to existing reconstruction-based methods Jiang et al. (2018); Zhuang et al. (2018); Chatzikonstantinou et al. (2020), our approach is data-free and computationally efficient, obviating the need for external data or time-consuming iterative processes. Extensive experiments on several benchmark datasets, including CIFAR-10 and ImageNet, demonstrate that our proposed method outperforms state-of-the-art pruning methods regarding both compression ratio and accuracy retention. Further experiments show that the trained MAE can transfer across different structures, even datasets, which suggests its probability to function as a universal pruning criterion.

In summary, our main contributions can be summarized as follows:

- **A new insight.** We build the bridge between pruning and self-supervised learning through masked image modeling on structured weights in networks. Based on the 'harder-reconstructed-more-important' assumption, we reveal that a well-trained MAE can be an efficient pruning criterion.
- **Efficient training pipeline.** To address the variable shapes of weight matrices from each layer, we propose an efficient training pipeline for MAE, enabling batch-wise, parallel training to enhance both its performance and stability. We visually illustrate the minor differences between reconstructed weights and their originals, demonstrating MAE's ability to uncover hidden semantic information. We further unveil the linear relationship between MAE's performance and the pruned model's accuracy, assisting us to further enhance algorithmic efficiency.
- **Pruning strategy exploration.** Based on the trained MAE, we explore diverse possible strategies to prune the network. By formulating pruning as a sample-without-replacement problem, it can balance between performance and computational efficacy.
- **Comprehensive experiments.** Experiments on CIFAR-10 and ImageNet datasets with various structures, including CNNs and ViTs, demonstrate that MAEP can achieve the best trade-off between accuracy and FLOPs compared with other state-of-the-art techniques. Furthermore, the trained MAE demonstrates transferability across different structures, even datasets, highlighting its potential as a versatile criterion.

## 2 RELATED WORK

### 2.1 EVALUATION METRICS FOR PRUNING

Structured pruning aims to identify unimportant structural components by an appropriately defined criterion. Existing pruning criteria can be broadly categorized into two types: hand-crafted heuristics-based and statistic-based. Handcrafted heuristics-based methods typically rely on predefined rules or metrics for identification, *e.g.*, the magnitude of weights or influence on the final loss. Norm-based pruning methods, which remove the filters with the smallest $\ell_1$-norm Li et al. (2016) or $\ell_2$-norm He et al. (2020b) by assuming that a smaller norm leads to less contribution to the final output, are among the most widely used. These methods assume that smaller norms lead to less contribution to the final output. In contrast, statistic-based methods, also known as data-driven pruning methods, leverage statistics derived from intermediate outputs or gradients to evaluate importance, thereby enabling more accurate and generalizable pruning decisions. For instance, the APoZ Hu et al. (2016) measures the fraction of input samples that a channel's activations are zero, which correlates well with the importance of the channels. Inspired by the discovery that the average rank of multiple feature maps generated by a single filter is always the same, HRank Lin et al. (2020) prunes filters with low-rank

feature maps based on the principle that low-rank feature maps contain less information. Although statistic-based methods may excel in generalizability, they usually require substantial additional computation and data to complete the evaluation process.

## 2.2 Reconstruction-based pruning

Reconstruction-based pruning leverages the concept of reconstruction, which typically involves calculating the reconstruction error to determine which filters are important for pruning. Among these methods, ThiNet Luo et al. (2017) is a groundbreaking approach that calculates the reconstruction error between the pruned and original networks. NRE Jiang et al. (2018) proposes layer-wise pruning by minimizing the reconstruction error in nonlinear units since the error can change significantly before and after activation. DCP Zhuang et al. (2018) proposes a reconstruction-based method similar to that of ThiNet, which considers the discriminating ability of each channel to determine its importance for pruning. Alternatively, HOS Chatzikonstantinou et al. (2020) employs an auxiliary reconstruction loss at the output of the network and intermediate layers for fine-tuning, rather than for guidance in pruning. In summary, reconstruction-based methods can accurately identify the importance of individual filters. However, these methods involve multiple forward and backward passes through the network for each iteration of pruning, making them computationally expensive.

## 2.3 Application of masked autoencoder

Masked autoencoder (MAE) He et al. (2021) is a self-supervised learning technique in computer vision that masks random patches of an input image and reconstructs the missing pixels. Various studies have explored MAE's application in different tasks over the years. For instance, VideoMAE Tong et al. (2022) proposed an efficient method for self-supervised video pre-training with customized video tube masking, which can extract more effective video representations during the pre-training process. Point-MAE Pang et al. (2022) investigated the performance of MAE in point cloud self-supervised learning, addressing the challenges posed by point cloud's properties, including leakage of location and uneven information density. Nevertheless, MultiMAE Bachmann et al. (2022) extends MAE to the multi-modal concept, where the training objective includes predicting multiple outputs besides the RGB image, ensuring tractability as well as cross-modality predictive coding. Our MAEP explores MAE's potential in pruning by viewing pruning as a self-supervised learning task, which can also be regarded as a special application of MAE.

# 3 The Proposed Method

## 3.1 Preliminaries

In this section, we formally introduce the symbols and notations.

**Masked Autoencoder.** Masked Autoencoder is a self-supervised learner that has an asymmetric encoder-decoder architecture, with an encoder that operates only on the visible patches (without mask tokens), along with a lightweight decoder that reconstructs the original image from the latent representation and mask tokens. Given an input image $x_i \in \mathbb{R}^{3 \times H \times W}$, it is first divided into non-overlapping patches. Then a subset of patches is sampled and the remaining ones are masked (i.e., removed). The remaining patches are first embedded by a MLP (MultiLayer Perceptron) and then processed via a series of Transformer blocks to obtain the tokens. The positional embeddings are added to all tokens to make them be aware of the location information in the image. Similarly, the decoder comprises another set of Transformer blocks and a linear projection that decodes the tokens to the pixel level. Then it is reshaped and reconstructs the input by predicting the pixel values for each masked patch.

**Structured Pruning.** A deep convolutional neural network (CNN) can be represented by $W_i \in \mathbb{R}^{N_{out}^i \times N_{in}^i \times s_i \times s_i}$ for $1 \leqslant i \leqslant L$, where $L$ denotes the total number of convolutional layers in the network. Here, $N_{out}^i$ and $N_{in}^i$ refer to the output channels and input channels, respectively, for the $i$-th convolution layer, while $s_i$ denotes the size of the convolution kernel. Assume the pruning rate of the $i$-th layer is $p_i$, the number of filters decreases from $N_{out}^i$ to $N_{out}^i \times (1 - p_i)$. Given a dataset $D$ with $N$ samples denoted as $D = \{x_i, y_i\}_{i=1}^N$ and a sparsity level $k$ (i.e., the total number of remaining filters in the pruned network). Structured pruning for CNNs aims to find a compact subset $W'$ from $W$ that minimizes the test loss, which can be formulated as:

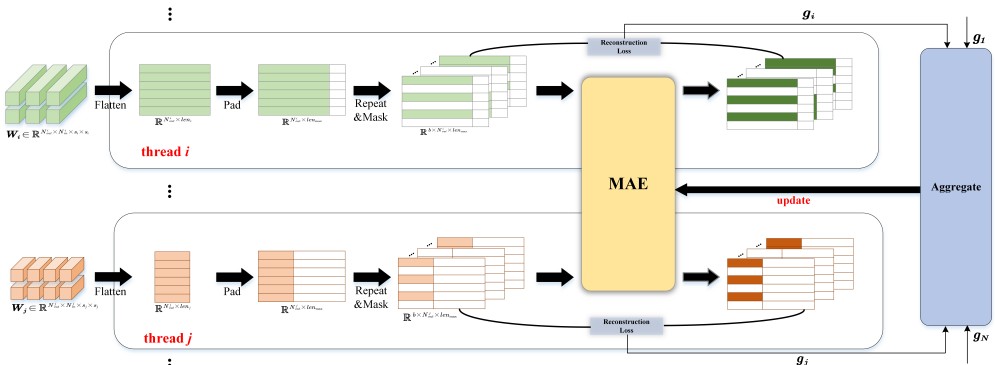

Figure 1: Workflow of the overall training pipeline. In each iteration, the weights from various convolutional layers are randomly assigned to each thread, where a sequence of operations is conducted to reconstruct the original weights. More specifically, filters from each layer are flattened, padded, and repeated to form a batch, which is then forward propagated with MAE for reconstruction to obtain the calculated gradient. Finally, gradients from each thread are aggregated to update MAE. (Best viewed in color.)

$$\min_{\boldsymbol{W}'} \ell\left(\boldsymbol{W}'; D\right) = \min_{\boldsymbol{W}'} \frac{1}{n} \sum_{i=1}^{n} \ell\left(\boldsymbol{W}'; (\boldsymbol{x_i}, y_i)\right), \qquad (1)$$
$$\text{s.t.} \quad \left\|\boldsymbol{W}'\right\|_0 \leqslant k.$$

where $\ell(.)$ is the loss function (*e.g.*, cross-entropy loss for image classification). $\boldsymbol{W}'$ represents the filter set of the pruned network. It's worth noting that while we use CNN as an example, the principles of structured pruning apply similarly to transformer-based architectures.

### 3.2 MAE AS AN IMPORTANCE INDICATOR

#### 3.2.1 EFFICIENT TRAINING PIPELINE OF MAE

In this subsection, we demonstrate the details of our proposed training pipeline, as depicted in Fig 1. Based on the reconstruction ability of MAE, our objective is to reconstruct filters within CNNs (or MLPs in the case of ViTs, with CNNs used here for ease of explanation). To accomplish this, we first flatten each convolutional filter in $\boldsymbol{W_i}$ to 2D dimension such that $\boldsymbol{W_i^f} \in \mathbb{R}^{N_{out}^i \times len_i}$, where $len_i = N_{in}^i \times s_i \times s_i$. Each row can be viewed as an image patch in the original setting of MAE. Then $m_i\%$ of the rows are masked, and the unmasked parts are sent to the MAE to obtain the reconstructed filter matrix.

Although it is possible to train a personal MAE for each convolutional layer, the total training cost is negligible. Besides, such personal MAE may overfit to a certain layer, therefore be biased toward it. However, the shape of the weight matrix from each layer is distinct, *e.g.*, the number of rows and columns are different. It is impossible to train a generalized MAE directly across layers. Therefore, we propose a suitable training pipeline to update our MAE. To first address the mismatch in columns, we pad each row in $\boldsymbol{W_i^f}$ with zeros to the maximum length $len_{max}$ in the network, where $len_{\max} = \max\{len_1, len_2, \dots, len_L\}$. Then we repeat the padded matrix for $b$ times and use different masks with the same mask ratio $m_i$ in each sample, where $b$ denotes the batch size and $\boldsymbol{W_i^{fm}} \in \mathbb{R}^{b \times N_{out}^i \times len_{\max}}$. This enables us to train in a batch-wise manner.

As $N_{in}^i$ increases when the network goes deeper, it becomes harder to reconstruct deeper layers' filters. Inspired by focal loss Lin et al. (2017), we regard filters from deeper layers as hard samples.

Figure 2: Visual comparison between the feature maps generated by the original filters (first row) from the 8th layer of ResNet-50 and those generated by the reconstructed filters (second row). Each image corresponds to a certain channel of the feature map. Comparing the slight variation between each image in both rows, it is confirmed that MAE can successfully learn the semantic information concealed within the filters.

Consequently, we adjust the terms in the reconstruction loss with balanced weights, giving more attention to hard samples and less to easy ones. Then the overall training loss can be formulated as:

$$L_{train} = \sum_{l=1}^{L} w_l{}^\gamma MSE\left(\boldsymbol{W_i^r}, \boldsymbol{W_i^{fm}}\right), \text{ where } w_l = \frac{len_l}{len_{\max}},$$

where $\gamma$ is a hyperparameter that balances each sample's term and $\boldsymbol{W_i^r}$ denotes the reconstructed filter matrix from MAE. It is worth mentioning that the MSE loss is only applied on the non-padding part of the masked rows, rather than the whole reconstructed matrix.

However, the samples in a batch come from the same convolutional layer, resulting in a lack of interactions between diverse layers due to the row mismatch issue mentioned earlier. To address this challenge, we opt to train the MAE in a distributed parallel manner, where the $i-th$ thread processes samples from a randomly sampled layer and aggregates the obtained gradients $g_i$ with other threads to update the MAE. The total number of threads is denoted as $N$, where each thread corresponds to one GPU in our experiments. With the proposed training pipeline, the training loss can decrease significantly, expediting convergence and enhancing stability. In our experiments, we also apply exponential moving average (EMA) to update the MAE during training.

In summary, the overall training process in one iteration for each thread can be described as follows:

---

1. Randomly select a convolutional layer in the target network and unflatten its weights to obtain $\boldsymbol{W_i^f}$
2. Pad $\boldsymbol{W_i^f}$ with zero to $len_{max}$ and forming a batch with different masks to get $\boldsymbol{W_i^{fm}}$.
3. Conduct forward propagation with MAE to get the reconstructed filters $\boldsymbol{W_i^r}$.
4. Calculate the reconstruction loss and aggregate the obtained gradients with other threads to update MAE.

---

To visually illustrate MAE's ability to capture knowledge within filters, we compare the generated feature maps from the reconstructed filters to the original filters in Fig. 2. It is observed that there is only a slight distinction between them, indirectly implying the minor difference between reconstructed and original filters. Given MAE's proficiency in reconstructing the masked filters, it is reasonable to conclude that MAE can effectively learn the semantic information concealed within the filters.

However, evaluating MAE's training efficacy via visualizing the feature maps can be cumbersome. Therefore, we introduce a quantitative evaluation criterion. Specifically, we calculate the total error of reconstructing every single filter in the network as the test loss for validation. As depicted in Fig. 3, we utilize MAE with different test losses to prune ResNet-56 on CIFAR-10 and record the pruned accuracy without fine-tuning. The figure illustrates that as the test loss decreases, the pruned accuracy exhibits an increasing trend with a Pearson Correlation coefficient of around $-0.90$. Given this strong linear correlation, we posit that it can serve as an indicator of MAE's training effectiveness, consequently resulting in a more finely pruned model.

### 3.2.2 EXPLORING DIVERSE PRUNING STRATEGIES

After training the MAE, we can prune the network based on the 'harder-reconstructed-more-important' assumption. Those structured parameters with higher reconstruction loss may contain some unique semantic information that cannot be reconstructed based on the hidden information in other parameters. In this section, we explore several strategies to locate the most-difficult-to-reconstruct parameters for preservation. Here, we take pruning CNNs as an example.

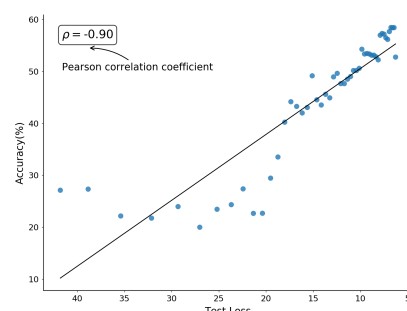

**Strategy 1: Exhaustive Search.** With a specified pruning rate, we can generate $N_m$ different masks and subsequently choose the one exhibiting the highest reconstruction error. Filters corresponding to the masked indices are the hardest to reconstruct, and they are therefore preserved. However, the total number of possible pruning combinations can be astronomically high. For example, for a convolutional layer with 256 filters, pruning 50% of it can account for more than $2^{100}$ possibilities. While this is the most accurate strategy, the associated computational demands render it impractical for broader and deeper CNN architectures.

Figure 3: Correlation between the test loss of MAE and the pruned accuracy, as observed through ResNet-56 experiments on the CIFAR-10 dataset. The trend shows that as the test loss reduces, the pruned accuracy increases, thus demonstrating that a more efficient MAE can be a better pruning guidance. The solid line is the result of fitting with `np.polyfit` for a better presentation.

**Strategy 2: Independent Selection.** To simplify the pruning process, we can evaluate the importance of each filter independently. Given the $i$-th convolutional layer, we mask each filter in turn and sort the reconstruction loss in descending order. Subsequently, we select and retain the top-k filters. However, this strategy disregards inter-filter relationships and may result in suboptimal solutions, despite its simplicity and efficiency.

**Strategy 3: Sample-Without-Replacement.** Suppose a fraction of filters have already been pruned, the difficulty of reconstructing the remaining parts differs from the initial stage. Independently evaluating filter importance can lead to significant performance degradation, especially at high pruning rates. Therefore, by taking advantage of the variable-length input characteristic of MAE, we formulate pruning as a sample-without-replacement problem, *e.g.*, always selecting the filter with the highest reconstruction loss from the remaining subset for preservation till the pre-defined sparsity. This strategy strikes a balance between the two prior ones, offering both superior computation efficiency and evaluation accuracy.

**Extension to Vision Transformers.** Apparently, our MAEP can also be applied to ViTs. Specifically, the training phase is essentially similar to that of CNNs, with the exception that 1) MLPs in vision transformers are inherently 2D matrices, eliminating the need for flattening 2) certain MLPs may consist of multiple attention heads and therefore need to be handled separately. Furthermore, as ViTs contain the unique self-attention module, the embedding dimension of the self-attention module must be the same across $W_Q/W_K/W_V$ and different heads to enable structured pruning of ViTs and gain realistic speed-up without customized hardware/software support. In our experiments, we use $W_K$ to determine the pruning of the self-attention module, considering the rich semantic information that is naturally embedded in it.

## 4 EXPERIMENTS

We perform several experiments on various datasets (CIFAR-10/ImageNet) and structures (CNNs/ViTs) to validate the effectiveness of MAEP. For all ImageNet experiments, we take the off-the-shelf model from either Pytorch or Huggingface Library. In the case of CIFAR-10, we train the networks with our own settings. All experiments are conducted on 8 V100 GPUs with Pytorch, where the detailed settings are provided in Appendix A.1.

Table 1: Pruning Results on CIFAR-10 dataset.

| Method | Base Acc.(%) | Acc↓(%) | FLOPs↓(%) | Method | Base Acc.(%) | Acc↓(%) | FLOPs↓(%) |
|---|---|---|---|---|---|---|---|
| *ResNet-56* | | | | *VGG-16* | | | |
| HRank Lin et al. (2020) | 93.30 | 0.13 | 50.0 | AutoPrune Xiao et al. (2019) | 92.40 | 0.90 | 23.0 |
| ASFP He et al. (2020b) | 93.59 | 1.15 | 52.6 | DeepPruningES Junior & Yen (2020) | 93.94 | 2.15 | 32.0 |
| CC Li et al. (2021) | 93.33 | 0.31 | 52.0 | VCNNP Zhao et al. (2019) | 93.25 | 0.07 | 39.1 |
| LFPC He et al. (2020a) | 93.59 | 0.35 | 52.9 | GAL Lin et al. (2019) | 93.96 | 3.23 | 45.2 |
| FWTW Elkerdawy et al. (2021) | 93.66 | 1.38 | 54.0 | CHIP Sui et al. (2021) | 93.96 | 0.10 | **58.1** |
| **MAEP (Ours)** | 93.59 | **0.02** | 55.9 | **MAEP (Ours)** | 93.87 | **0.06** | 57.6 |
| *ResNet-110* | | | | *DenseNet-40* | | | |
| ASFP He et al. (2020b) | 93.68 | 0.48 | 52.3 | SOSP Nonnenmacher et al. (2021) | 94.58 | **0.35** | 38.8 |
| HRank Lin et al. (2020) | 93.50 | 0.14 | 58.2 | HRank Lin et al. (2020) | 94.81 | 0.57 | 40.8 |
| LFPC He et al. (2020a) | 93.68 | 0.61 | 60.3 | GAL Lin et al. (2019) | 94.81 | 1.28 | 54.7 |
| DECORE Alwani et al. (2021) | 93.50 | 0.00 | **61.8** | DECORE Alwani et al. (2021) | 94.81 | 0.77 | 54.7 |
| **MAEP(Ours)** | 93.90 | **-0.06** | 59.3 | **MAEP (Ours)** | 94.88 | 0.56 | **55.3** |

Table 2: Pruning Results on ImageNet dataset.

| Method | Base Top-1(%) | Pruned Top-1(%) | Base Top-5(%) | Pruned Top-5(%) | Top-1↓(%) | Top-5↓(%) | FLOPs↓(%) |
|---|---|---|---|---|---|---|---|
| *ResNet-18* | | | | | | | |
| CHEX Hou et al. (2022) | 70.30 | 69.60 | - | - | **0.70** | - | 42.3 |
| PFP Liebenwein et al. (2020) | 69.74 | 65.65 | 89.07 | 86.75 | 4.09 | 2.32 | 43.1 |
| SCOP Tang et al. (2020) | 69.74 | 68.62 | 89.08 | 88.45 | 1.14 | **0.63** | 45.0 |
| HFP Enderich et al. (2021) | 69.75 | 68.53 | - | - | 1.22 | - | 45.0 |
| **MAEP (Ours)** | 69.75 | 68.91 | 89.07 | 88.34 | 0.94 | 0.73 | **45.9** |
| *ResNet-50* | | | | | | | |
| CHEX Hou et al. (2022) | 77.80 | 77.40 | - | - | 0.40 | - | 51.3 |
| PaS Li et al. (2022) | 77.10 | 76.70 | 93.50 | 93.10 | 0.40 | 0.40 | 51.3 |
| Polarize Zhuang et al. (2020) | 76.15 | 75.63 | - | - | 0.52 | - | 54.0 |
| SCOP Tang et al. (2020) | 76.15 | 75.26 | 92.87 | 92.53 | 0.89 | 0.34 | 54.6 |
| **MAEP (Ours)** | 76.15 | 76.10 | 92.87 | 92.73 | **0.05** | **0.14** | **60.5** |
| *DeiT-Tiny* | | | | | | | |
| $S^2$ViTE Chen et al. (2021) | 72.20 | 70.12 | 91.10 | - | 2.08 | - | 23.7 |
| GOHSP Yin et al. (2023) | 72.20 | 70.24 | 91.10 | - | 1.96 | - | 30.0 |
| SAViT Zheng et al. (2022) | 72.20 | 70.72 | 91.10 | - | 1.48 | - | 24.4 |
| WDPruning Yu et al. (2022) | 72.20 | 71.10 | 91.10 | 90.09 | 1.10 | 1.01 | 30.8 |
| **MAEP (Ours)** | 72.20 | 72.61 | 91.10 | 91.28 | **-0.41** | **-0.18** | **38.5** |
| *DeiT-Small* | | | | | | | |
| WDPruning Yu et al. (2022) | 79.90 | 78.55 | 95.00 | 94.37 | 1.25 | 0.63 | 32.6 |
| $S^2$ViTE Chen et al. (2021) | 79.90 | 79.22 | 95.00 | - | 0.68 | - | 31.6 |
| SAViT Zheng et al. (2022) | 79.90 | 80.11 | 95.00 | - | -0.26 | - | 31.7 |
| GOHSP Yin et al. (2023) | 79.90 | 79.98 | 95.00 | - | -0.08 | - | 35.0 |
| **MAEP (Ours)** | 79.90 | 80.40 | 95.00 | 95.52 | **-0.50** | **-0.52** | **36.3** |
| *Swin-Tiny* | | | | | | | |
| ViT-Slim Chavan et al. (2022) | 81.20 | 80.70 | 95.50 | - | 0.50 | - | 24.4 |
| X-Pruner Yu & Xiang (2023) | 81.20 | 80.70 | 95.50 | - | 0.50 | - | 28.9 |
| **MAEP (Ours)** | 81.20 | 80.99 | 95.50 | 95.22 | **0.21** | **0.28** | **33.4** |

## 4.1 COMPARISON ON CIFAR-10

We present the results of CNNs on CIFAR-10, which are summarized in Table 1. In most cases, our method achieves a larger reduction in FLOPs or more competitive performance when compared to other state-of-the-art pruning methods. For example, our method accelerates ResNet-56 by 55.9% with an only 0.02% accuracy drop, surpassing FWTW, which can only prune 54.0% of the total FLOPs but incurs a 1.38%accuracy drop. These results show that irrespective of the network architecture, our MAEP produces a compact structure with competitive complexity.

## 4.2 COMPARISON ON IMAGENET

We further prune ResNet-18/50, DeiT-Tiny/Small and Swin-Tiny models with MAEP and present a comparison with other contemporary methods on the challenging ImageNet dataset. The results are summarized in Table 2. MAEP generally outperforms its counterparts across various aspects, encompassing top-1 and top-5 accuracy as well as FLOPs reduction. Specifically, MAEP surpasses SCOP, which achieves only 75.26% top-1 accuracy and a 54.6% reduction in FLOPs, both inferior to that of MAEP. For ViT, our method achieves a 38.5% reduction in FLOPs on DeiT-Tiny, which is larger than that of WDPruning, with a lower top-1 drop (-0.41% v.s. 1.10%) and top-5 drop (-0.18% v.s. 1.01%). Moreover, this superiority extends to the larger Deit-Small model, where MAEP

Table 3: Transferability of the trained MAE. Weights from each source model are utilized to train an MAE, which is then applied to prune other target models.

| Source \ Target | ResNet-56 | VGG-16 | ResNet-18 | ResNet-50 |
|---|---|---|---|---|
| ResNet-56 | 93.60 | **93.34** | 67.96 | 75.11 |
| VGG-16 | 93.62 | 93.17 | 68.06 | 75.32 |
| ResNet-18 | **93.72** | 93.05 | **68.18** | 75.48 |
| ResNet-50 | 93.66 | 93.21 | 68.12 | **75.78** |

consistently outperforms alternative methods to a significant degree. For further results of actual speedup, please refer to Table 5

### 4.3 TRANSFERABILITY OF THE TRAINED MAE

While we avoid the time-consuming training process for each convolutional layer's MAE, it remains impractical to train a unique MAE for every unpruned network. Remarkably, we observe that a well-trained MAE exhibits transferability across different depths, architectures, and even datasets, as evidenced by Table 3. To be specific, we train an individual MAE on each source model and then utilize it to compress various target models. We set the compression rate to 50% for all cases. We represent each component of the table as (#source, #target). In cases where the $len_{max}$ of the source model is smaller than that of the target model, we reinitialize the projection layer in both the encoder and decoder and only fine-tune it to ensure dimension alignment between source and target. The results of (ResNet-18, ResNet-50) and (ResNet-50, ResNet-18) reveal the MAE's ability to transfer across depths, whether from deeper to shallower networks or vice versa. Furthermore, (ResNet-56, VGG-16) achieves an accuracy of 93.34%, underscoring the MAE's capability to transfer across different architectural structures. Take ResNet-56 as the target, both (ResNet-18, ResNet-56) and (ResNet-50, ResNet-56) achieve competitive results. This showcases the MAE's potential to transfer from complex datasets to simpler ones with identical architectural structures. Interestingly, the best results of ResNet-56 are obtained when ResNet-18 is used as the source rather than ResNet-56 itself. We speculate that this discrepancy may be correlated with the overfitting during the MAE training on ResNet-56. Moreover, both (ResNet-56, ResNet-50) and (VGG-16, ResNet-50) yield satisfactory outcomes. However, due to the limited representative ability of filters from lighter models, there seems to exist a domain gap that hindering from optimal performance. In conclusion, owing to this transferability, we contend that there is no necessity to train a unique MAE for every unpruned network. Instead, a well-trained MAE can be regarded as a universal pruning criterion.

### 4.4 ABLATION STUDIES

**Different pruning strategies.** Fig. 4(a) shows the results of various pruning strategies when pruning ResNet-56 on CIFAR-10, with pruning rates ranging from 10% to 90%. As the rate of pruning increases, the fine-tuned accuracy of all strategies decreases. Still, we can observe that **Strategy 3** consistently outperforms **Strategy 2** in all cases. This superiority stems from its consideration of inter-filter correlations by formulating pruning as a sample-without-replacement problem. Moreover, **Strategy 1** can yield superior performance when $N_m$ is large enough. In our experiment, $N_m = 5000$ provided satisfactory results for ResNet-56. However, a decrease in $N_m$ to 100 resulted in a significant drop in accuracy. Fig. 4(b) provides additional insight into the computational complexity of various strategies across layers with a 50% pruning rate. As the layer goes deeper, all strategies show an upward trend of FLOPs required for pruning. The FLOPs occupation of **Strategy 2** consistently falls between that of the other two strategies, indicating it can provide a better computational efficacy. For more complex networks like ResNet-50, a large value of $N_m$ imposes a heavy computational burden, making it impractical for efficient deployment.

**Variation in hyperparameter $\gamma$.** The balancing between hard and easy samples is controlled by coefficients $\gamma$. We vary $\gamma$ between 0, 1, 5, and 10 to explore its impact on the MAE. Fig. 4(b) shows the impact on the test loss, with MAE trained on ResNet-56. Compared to $\gamma = 0$, the introduction of a balanced weighting factor into MAE's training ($\gamma = 1$) leads to a reduction in the test loss. Given the linear correlation between MAE's test loss and the pruned model's accuracy, a well-trained MAE offers a more precise evaluation of filter importance. However, as $\gamma$ grows to 5, hard samples dominate the initial training stage, resulting in a narrowing loss term on easy samples, which inevitably slows down the convergence speed. As $\gamma$ further increases, the test loss fails to reach the desired scope within the given epochs, indicating that $\gamma$ must be carefully selected for proper balance.

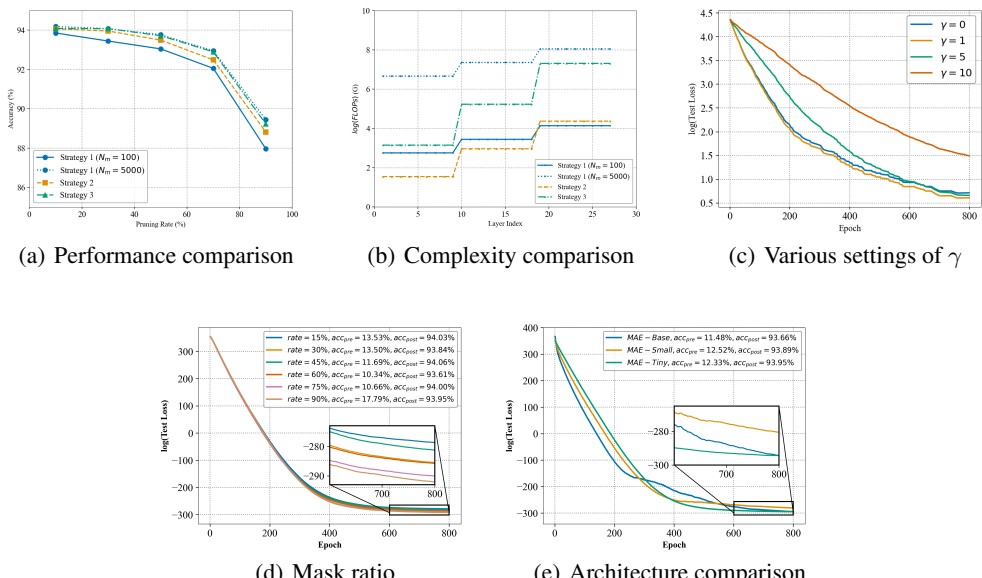

(a) Performance comparison    (b) Complexity comparison    (c) Various settings of $\gamma$

(d) Mask ratio    (e) Architecture comparison

Figure 4: (a) Accuracy and (b) complexity comparison of different pruning strategies of ResNet-56 on CIFAR-10 dataset. (c) Test loss of MAE with different values of $\gamma$. (d) Comparison between different mask ratios of MAE. (e) Architectural choices of MAE.

**Mask ratio of MAE.** According to the original MAE He et al. (2021), the mask ratio seriously influences the final performance of the model. Therefore, we conduct a systematic investigation into the impact of it in our approach, and the results are presented in Fig. 4(d). We vary it from 15∼90% when pruning ResNet-56 on Cifar-10. From the curves of the test loss, it can be seen that the MAE under different mask ratios can all converge, with no significant gap in the final loss. Moreover, from the accuracy before (denote as $acc_{pre}$) and after ($acc_{post}$) fine-tuning, it is evident that all variants of MAE effectively identify and filter out unimportant parameters. We hypothesize that MAE is insensitive to mask ratios due to its robust generalization ability and the fact that it does not need to perfectly reconstruct the parameters to discriminate the importance between them.

**Impact of MAE Architecture.** To verify whether the architecture of MAE affects the performance of the pruned model, we conduct a series of ablation experiments, where the results are depicted in Fig. 4(e). Adhering to the design principles of MAE, we proceed to reconstruct the parameters of ResNet-56 using MAE-Base, MAE-Small, and MAE-Tiny, respectively. The training curves exhibit not much distinctions across various architectures, except that MAE-Base's training is more unstable and tends to slightly overfitting. In terms of post-pruning accuracy, all variants perform similarly, implying that even the most lightweight MAE-Tiny can serve well as an importance indicator.

## 5    CONCLUSION

We propose a novel structured pruning method, MAEP, to obtain a compact structure of deep neural networks. MAEP brings new insights into the field by building the bridge between pruning and self-supervised learning, shedding light on the potential of MAE as an efficient importance indicator. MAEP (1) proposes an efficient training pipeline, which expedites convergence and enhances stability; (2) proves that MAE can uncover the semantic features both visually and quantitatively through masked image modeling, indicating a better-trained MAE can lead to a better-pruned model; (3) explores diverse pruning strategies and strike a balance between performance and algorithm complexity by formulating it as a sample-without-replacement problem. Experiments on multiple datasets and structures demonstrate that MAEP can effectively reduce the FLOPs of CNNs/ViTs while achieving superior accuracy compared to other state-of-the-art methods. Ablation studies further provide additional evidence of the trained MAE's capacity to transfer across different network structures and datasets, highlighting its potential as a universal pruning criterion.

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

# A  APPENDIX

## A.1  EXPERIMENTAL SETTINGS

**Datasets.** The proposed MAEP is evaluated through empirical experiments on two widely used image classification datasets, CIFAR-10 Krizhevsky (2009) and ImageNet Deng et al. (2009), to demonstrate its effectiveness. The CIFAR-10 dataset consists of 60,000 32x32 images with 10 classes, where 50,000 images are used for training, and the remaining 10,000 images are used for testing. On the other hand, the large-scale ImageNet dataset includes 1.28 million 224x224 training images and 50,000 validation images drawn from 1,000 categories. For both datasets, the data was preprocessed by subtracting the mean and dividing the standard deviation, and the same data augmentation approach as He et al. (2020b) is used.

**Model Architectures.** To provide a fair comparison with other methods, the evaluation of our method is performed on various mainstream CNN and ViT models. For CNNs, it includes models like VGGNet Simonyan & Zisserman (2014), ResNet series He et al. (2015), and DenseNet Huang et al. (2016) For ViTs, the method is tested on DeiT Touvron et al. (2021) and SWIN Liu et al. (2021). For the default structure of MAE, we set the output dim of the encoder and decoder to 768 and 512, respectively. The depth of the encoder and decoder are 12 and 8 with 64 as the dimension of each head.

**Implementation Details.** We follow the traditional train-prune-finetune pipeline in most of the pruning methods. To train the baseline networks, we use the popular stochastic gradient descent

(SGD) optimizer with momentum. For the CIFAR-10 dataset, we train the networks using a mini-batch size of 256 for 200 epochs with a momentum of 0.9, starting with an initial learning rate of 0.1. On the other hand, for the ImageNet dataset, we choose the pre-trained weights from Pytorch Library. The MAE is trained for 8000 epochs with a batchsize of 256, using AdamW Loshchilov & Hutter (2017) optimizer with an initial learning rate $1.5 \times 10^{-4}$. The hyperparameter $\gamma$ is set to 1 by default. We prune along the output dimension of convolutional layers in CNNs and FFNs in ViTs. For pruning ViT, we also prune the hidden dimension of the self-attention module (*e.g.* $W_Q/W_K/W_V$). Due to limited computational resources, we set the pruning rate of different components to be the same and do not tune it additionally. Theoretically, a proper adjustment can further motivate the potential of MAEP. During fine-tuning of the pruned model, we adopt the warm-up strategy and cosine annealing strategy to adjust the learning rate. For the CIFAR-10 dataset, we finetune the pruned model for 400 epochs, while on ImageNet, the number of finetuning epochs is 120 for CNNs and 300 for transformers (the same setting on which they are originally pre-trained). To ensure the reproducibility of our results, we conduct each experiment three times and report the mean value for comparison.

## A.2 MORE EXPERIMENTAL RESULTS

### A.2.1 PRUNING SELF-ATTENTION MODULE

In contrast to CNNs, in which convolutional layers are the dominant component, there exists the unique self-attention module in ViTs. By default, we leverage the inherent semantic information in $W_K$ to guide the pruning of self-attention. However, we also measure the impact of using different parts as anchors for pruning half of the dimension of self-attention in DeiT-Tiny, where the results are summarized in Table 4. In terms of the results, $W_K$ is slightly better than the other settings. Although the pre-finetuning accuracy of taking average is slightly higher, it eventually does not show a consistent advantage. It is also worth mentioning that the results of concatenating the weights of self-attention are identical to using $W_Q$. Our hypothesis is that this occurs because $W_Q$ is more challenging to reconstruct compared to the other components, making it dominant in the pruning process.

| feature | $acc_{pre}(\%)$ | $acc_{post}(\%)$ |
|---|---|---|
| $W_Q$ | 1.00 | 73.98 |
| $W_K$ | 1.25 | **74.20** |
| $W_V$ | 1.07 | 74.07 |
| Concat | 1.00 | 73.98 |
| Mean | **1.40** | 74.01 |

Table 4: Impact of choosing different parts to prune the self-attention module with the pruning ratio of it is set to 50%. $W_K$ shows the best results over others.

Table 5: Measurement of inference speedup (images per second) of models on ImageNet dataset. The experiments are carried out on Nvidia RTX4090 GPU with 256 as the batchsize.

| Model | FLOPs↓(%) | Speedup | Top-1 Acc. (%) |
|---|---|---|---|
| | 30.8 | 1.22× | 73.53 |
| DeiT-Tiny | 38.5 | 1.34× | 72.61 |
| | 46.2 | 1.38× | 71.70 |
| DeiT-Small | 36.3 | 1.29× | 80.40 |
| Swin-Tiny | 33.4 | 1.19× | 80.99 |
| ResNet-50 | 60.5 | 1.27× | 76.10 |

### A.2.2 THROUGHPUT MEASUREMENT

Considering that FLOPs only reflect the complexity of the model and are not sufficiently objective, we further measured the throughput of the pruned model. We set the batchsize to 256 and measure it on Nvidia RTX4090 GPUs, where the experimental results are summarized in Table 5. For example, while reducing FLOPs by 38.5%, MAEP enables DeiT-Tiny to gain 72.61% top-1 accuracy with 1.34× throughput improvement. However, the speedup on ResNet-50 is not satisfactory, presumably owing to the repeated dimension transformations with excessive residual connections that inhibit the inference speedup.

