# OpenReview forum: "Pruning-as-Reconstruct: Masked Autoencoders are Efficient Importance Indicators"
_ICLR.cc/2024/Conference — ICLR 2024 Conference Withdrawn Submission_

### Official Review · Reviewer_GDLV · 2023-10-29

**Soundness:** 2 fair
**Presentation:** 2 fair
**Contribution:** 2 fair
**Rating:** 3
**Confidence:** 4

**Summary:**

This study shows that masked autoencoders (MAEs) can be used to compute weight importance in structured pruning. Based on the ‘harder-reconstructed-more-important’ assumption, the convolutional filters in CNNs (or the linear projection matrices in ViTs) work as the training data of MAEs. The weights that are challenging for the MAE to reconstruct are deemed vital and remain unpruned.

**Strengths:**

- The idea of reconstructing weights parameters under the MAE framework seems new.
- The applicability is shown not only for CNNs but also for ViTs, and the transferability of trained MAEs is demonstrated.

**Weaknesses:**

- It's unclear why MAE was chosen as the primary method for determining importance. I think the core of this framework is capturing inter-weight relationships/correlations via a reconstruction task, which implies that other networks can be also workable. What distinct benefits do MAEs offer? Is it possible to design other baselines (e.g., vanilla AE without masking) and conduct the comparison?
- Considering the added complexity of the proposed method, such as the requirement for individual MAE training, the performance shown in Tables 1 and 2 hardly seems worthwhile. Additionally, the ResNet baselines for ImageNet experiments look outdated (works in 2020, 2021) or hard to understand (works in 2022 with different Base Top-1).
- It is difficult to understand where the performance gains come from, compared to the baseline methods. In particular, in Figure 4(d) and (e), better post-training accuracies (without finetuning) did not guarantee better final accuracies (with finetuning).
- Please report the actual throughput and/or latency, which is the main advantage of structured pruning.
- Analyzing which parts were pruned would be interesting to understand the network behavior.
- Is it possible to show the transferability of trained MAEs for the ViT experiments? I think it would be beneficial to claim “a well-trained MAE can be regarded as a universal pruning criterion.”

**Questions:**

Described in the above Weaknesses section

---

### Official Review · Reviewer_XNx5 · 2023-10-30

**Soundness:** 2 fair
**Presentation:** 2 fair
**Contribution:** 2 fair
**Rating:** 3
**Confidence:** 4

**Summary:**

The paper proposes a masked autoencoder (MAE) for structured pruning. MAE conducts pruning based on reconstruction criteria and self-learning schema. Numerical experiments validate the efficacy of the proposed method.

**Strengths:**

- The paper is written well in general and easy to follow.
- The reconstruction and self-learning for pruning criteria make senses, that should lead to better performance.

**Weaknesses:**

- **Some statements are not correct.**  For example, in introduction **existing pruning criteria are
 predominantly based on handcrafted heuristics or calculated statistics, hindering
 their generality and effectiveness.** In fact, many pruning methods design the pruning criteria based on numerical optimization, which are generic for general applications.
- **The pruning method is not that appealing.**  The current trend of structured pruning is automated and ease-to-use, such as Torch-Pruning (DepGraph) and OTOv2. The proposed MAE seems computational expensive, hard to use, and a case-by-case approach, which is opposite to the main trend. Meanwhile the paper does not discuss and compare with these automated pruning approaches.

- **The novelty is limited.** Reconstruction for pruning has been proposed. The combination with self-learning seems not that dedicately designed and not well presented.

**Questions:**

See the weakness.

---

### Official Review · Reviewer_Q6JX · 2023-11-01

**Soundness:** 3 good
**Presentation:** 3 good
**Contribution:** 3 good
**Rating:** 5
**Confidence:** 3

**Summary:**

This paper seeks to construct a feasible criterion that can provide importance evaluation on network pruning without explicit reliance on quantitative data. To this end, the authors propose a well-trained pruning criterion based on the masked autoencoder, to fully exploit the intrinsic correlations among learned parameters.

**Strengths:**

1. The idea of constructing a universal and learnable pruning criterion to access the pruned accuracy is reasonable.
2. This paper brings new insights into the field by building the bridge between pruning and self-supervised learning.

**Weaknesses:**

1. The minor difference between reconstructed and original filters in Figure 2 is insufficient to demonstrate that MAE can learn the semantic information hidden in the filters.
2. Observations through ResNet-56 experiments on the CIFAR-10 dataset is insufficient to establish a clear relationship between the performance of MAE and the pruned accuracy. The authors should validate the assumption of "harder-reconstructed-more-important" on more models and datasets.
3. Could you analyze the cases in Table 1 where the proposed MAEP is inferior to the baselines? Please provide more "inference speedup" results in Tables 1 and 2.
4. Section 4.3 states that "a well-trained MAE can be regarded as a universal pruning criterion". However, the conclusion lacks credibility as Table 3 does not show a universal MAE that performs the best on all target models.
5. The ablation experiment of Figure 4(d) only relies on pruning ResNet-56 on CIFAR-10, which is insufficient to prove that MAE is insensitive to mask ratios. More experimental results should be provided.
6. The fonts in all figures and tables in the paper are too small to read, especially in Figure 1. There are also grammar and punctuation errors in the paper. Please correct them carefully.

**Questions:**

Please refer to the weakness part.

---

### Official Review · Reviewer_WXme · 2023-11-01

**Soundness:** 3 good
**Presentation:** 3 good
**Contribution:** 3 good
**Rating:** 5
**Confidence:** 5

**Summary:**

In this work, the authors propose a new structured pruning method (MAEP) by using masked autoencoders as indicators of weight importance. The proposed MAEP mainly includes two steps. In the first step, the authors reconstruct the weight of the networks by using masked autoencoders. To make the training efficient, the authors fix the length of the weight matrix and balance the reconstruction loss for different layers.  In the second step, the authors prune the network with the assumption of "harder-reconstructed-more-important" by using the proposed Sampe-Without-Replacement strategy for the reconstructed parameters.  In the experiments, the authors conduct experiments with prior pruning works and show better performance on multiple benchmarks with varied network structures.

**Strengths:**

1. The proposed MAEP is interesting and novel which prunes the weights according to their reconstruction difficulty by using masked autoencoders.

2. The writing is well and easy to follow.

3. Obtain better accuracy compared to the pruning methods.

**Weaknesses:**

1. The training cost of the proposed method is unknown compared to the conventional pruning method.

2. It's unclear how to obtain the reported accuracy after pruning.

3. Regarding MAE, the authors propose two techniques: 1) fix the row in the weight matrix as the maximum length in a network; 2)  adjust the reconstruction loss for different layers. Both methods are not related to "efficiency" which involves extra computation cost, but serve to enable parameter modeling by using MAE

**Questions:**

The detailed questions regarding Weaknesses.

1. The training cost of the proposed method is unknown compared to the conventional pruning method.

    1.1 What is the training cost of MAE compared to conventional supervised pruning methods? This comparison helps to justify the efficiency of the proposed method.

2.  It's unclear how to obtain the reported accuracy after pruning.

     2.1 What is the initial network? Is it a well-trained network or a random initialized model?

     2.2 The proposed pruning strategy directly removes unimportant weights without any supervision or weight update. Does the pruned model need to be further fine-tuned by using supervised learning to obtain the reported accuracy?

3.  It's better to provide more intuition to justify the "harder-reconstructed-more-important" assumption for purning